# Trends of HIV Mortality between 2001 and 2018: An Observational Analysis

**DOI:** 10.3390/tropicalmed6040173

**Published:** 2021-09-24

**Authors:** Chinmay Jani, Kripa Patel, Alexander Walker, Harpreet Singh, Omar Al Omari, Conor Crowley, Dominic C. Marshall, Richard Goodall, Arashdeep Rupal, Justin D. Salciccioli, Joseph Shalhoub

**Affiliations:** 1Department of Medicine, Mount Auburn Hospital/Beth Israel Lahey Health, Cambridge, MA 02138, USA; alexander.walker@mah.harvard.edu (A.W.); omar.alomari@mah.harvard.edu (O.A.O.); arashdeep.rupal@mah.harvard.edu (A.R.); 2Department of Medicine, Harvard Medical School, Boston, MA 02115, USA; justin.salciccioli@gmail.com; 3Medical Data Research Collaborative, London W2 1NY, UK; drhpsingh101@gmail.com (H.S.); crowconor@gmail.com (C.C.); dominic.marshall@bath.edu (D.C.M.); r.goodall@imperial.ac.uk (R.G.); j.shalhoub@imperial.ac.uk (J.S.); 4Smt NHL Municipal Medical College, Ahmedabad 380006, Gujarat, India; kripa2493@gmail.com; 5Division of Pulmonary and Critical Care, Medical College of Wisconsin, Milwaukee, WI 53226, USA; 6Division of Pulmonary and Critical Care, Lahey Hospital, Burlington, MA 01805, USA; 7National Heart and Lung Institute, Imperial College London, London SW3 6LY, UK; 8Department of Surgery and Cancer, Imperial College London, London SW7 5NH, UK; 9Division of Pulmonary and Critical Care, Brigham and Women’s Hospital, Boston, MA 02115, USA; 10Imperial Vascular Unit, Imperial College Healthcare NHS Trust, London W2 1NY, UK

**Keywords:** HIV, mortality, WHO, world

## Abstract

Since the beginning of the epidemic in the early 1980s, HIV-related illnesses have led to the deaths of over 32.7 million individuals. The objective of this study was to describe current mortality rates for HIV through an observational analysis of HIV mortality data from 2001 to 2018 from the World Health Organization (WHO) Mortality Database. We computed age-standardized death rates (ASDRs) per 100,000 people using the World Standard Population. We plotted trends using locally weighted scatterplot smoothing (LOWESS). Data for females were available for 42 countries. In total, 31/48 (64.60%) and 25/42 (59.52%) countries showed decreases in mortality in males and females, respectively. South Africa had the highest ASDRs for both males (467.7/100,000) and females (391.1/100,000). The lowest mortalities were noted in Egypt for males (0.2/100,000) and in Japan for females (0.01/100,000). Kyrgyzstan had the greatest increase in mortality for males (+6998.6%). Estonia had the greatest increase in mortality for females (+5877.56%). The disparity between Egypt (the lowest) and South Africa (the highest) was 3042-fold for males. Between Japan and South Africa, the disparity was 43,454-fold for females. Although there was a decrease in mortality attributed to HIV among most of the countries studied, a rising trend remained in a number of developing countries.

## 1. Introduction

In 2019, around 38 million individuals were living with human immunodeficiency virus (HIV), and 690,000 died due to HIV-related illness. Since the beginning of the epidemic in the early 1980s, HIV-related illnesses have led to the deaths of over 32.7 million individuals [1]. However, there were approximately 39% fewer deaths due to HIV in 2019 than in 2010 [2,3,4].

Overall decreases in HIV-related mortality have mostly been attributed to the significant improvements in HIV/AIDS management, including highly active antiretroviral combination therapies [5,6]. Newer medications have better side effect profiles as well, which has improved adherence [7,8]. Highly Active Antiretroviral Therapy (HAART) has also contributed to the increased life expectancy of patients with HIV [7]. Along with the Joint United Nations Program on HIV/AIDS (UNAIDS), the WHO has framed various policies and guidelines to decrease the global burden of morbidity and mortality due to HIV [9]. Strategies were developed throughout the world to address the needs of high-risk populations and increase their awareness [10,11]. These multifactorial efforts have led to an overall decrease in HIV mortality worldwide; however, certain areas have continued to see rising mortality rates. Active efforts are needed in those areas [2].

In 2014, UNAIDS and partners set goals they named the 90-90-90 targets. These goals were to diagnose 90% of all HIV-positive people, to provide antiretroviral therapy (ART) for 90% of those diagnosed, and to achieve viral suppression for 90% of those treated by 2020 [12,13]. This led to more widespread and earlier administration of ART. It was thus important to examine how the setting of these targets—and the subsequent efforts to meet them—impacted global HIV mortality trends.

The primary objective of this study was to describe current ASDRs for HIV. The secondary objectives were to compute the absolute and percentage changes in ASDRs, categorized by country and gender, through an observational analysis of HIV mortality data from 2001 to 2018 from the World Health Organization (WHO) Mortality Database. We also compared HIV mortality trends in 48 nations between 2001 and 2018.

## 2. Materials and Methods

### 2.1. Data Sources

We extracted HIV mortality data from the WHO Mortality Database from 2001 to 2018 for the member nations with available data in October 2020. Countries were divided based on their WHO specified regions. Data were extracted based upon the International Classification of Diseases-10th Edition (ICD-10) diagnosis codes B20, B21, B22, B23, B24, and R75 for HIV. The WHO evaluated the quality of these data to ensure comparability and reliability, without adjustment for underreporting. The proportion of all deaths registered in the population covered by the vital registration system for a given country (referred to as completeness) were estimated by the WHO for the latest available year. For member states with incomplete vital registration systems, demographic techniques were used by the WHO to estimate the level of completeness of death recording for the specified population, in order to allow for the calculation of the death rate [14]. For inclusion criteria, we first evaluated the database to determine the countries with available data. Out of 194 member countries of the WHO, data on HIV mortality were available for 118 countries. Based on the WHO mortality database 2009–2017 completeness data, 107 countries with >20% completeness were included in our data review. Out of these 107 countries, we further excluded 59 countries that either did not have data for five years or had significant data breaks, defined as greater than three consecutive years.

Crude mortality rates were dichotomized by gender and reported by year. We computed age-standardized death rates (ASDRs) per 100,000 people using the world standard population. The ASDR was defined as mortality, weighted to the distribution of mortality per 5-year age group, according to the world standard population [15]. This removed the effects of historical events on age structure and controlled for differences in age structure in populations, producing age-specific mortality rates and more representative data. The estimated level of coverage for deaths with a recorded cause for death was calculated by dividing the total deaths reported for a country in a given year from the vital registration system by the total estimated deaths for that year for its national population. The national population estimates used were those of the UN Population Division. Best estimates of death rates by age and sex (adjusted for incompleteness and incomplete coverage) were applied to the national population data to obtain total estimated deaths. The WHO estimated coverage for a member state may have been less than 100% due to incompleteness of registration, coverage of only some parts of the national population, or differences between the vital registration the UN estimated de facto population. Population and birth recording in all countries are specified in the data as per the WHO standard for inclusion in the database [14].

### 2.2. Statistical Analyses

We computed male and female mortality rates and used a locally weighted scatterplot smoothing (LOWESS) plot to fit the rates of male and female mortality using SAS v9.4 (SAS, Cary, NC, USA) (Figure 1). LOWESS was only used to make plots to visualize country-specific trends and not to model changes across the countries. Mortality data were missing in a small subset of countries in the WHO Mortality Database for one or more calendar years. We excluded countries with missing data of more than three consecutive years during the observation period. There were no other modifications to the data. Changes in ASDR over the observation period are calculated as crude absolute differences between first and last data points for the earliest and most recent years available. Similar to our previous studies, percentage change was calculated as [(End ASDR − Start ASDR)/Start ASDR] ∗ 100 for each gender and country [16,17]. We have also provided absolute changes in the ASDR. This can serve as a comparator to the percentage changes, especially for the countries with low baseline mortality.

## 3. Results

We analyzed data from 48 countries from the following regions (as classified by the WHO): Americas, Western Pacific, South East Asia, Europe, Eastern Mediterranean, and Africa. Amongst 48 countries, data for females were available for only 42 countries. Of 48 countries, 4 countries had data available until 2018, 18 until 2017, 12 until 2016, 7 until 2015, 4 until 2014, 1 until 2013, 1 until 2007, and 1 until 2005. Region-wise, the Americas included the USA and Canada. Europe included Armenia, Austria, Belgium, Croatia, Cyprus, Czech Republic, Denmark, Estonia, Finland, France, Georgia, Germany, Hungary, Ireland, Israel, Italy, Kyrgyzstan, Latvia, Lithuania, Luxembourg, Malta, Netherlands, Norway, Poland, Republic of Moldova, Romania, Serbia, Slovakia, Slovenia, Spain, Sweden, Switzerland, and the Turkey, United Kingdom. The Western Pacific region included Japan, Malaysia, Singapore, Australia, and New Zealand. The Eastern Mediterranean region included Egypt, Bahrain, Jordan, and Kuwait. Thailand was the only country included in the South East Asian region. The African region included the countries of Mauritius and South Africa.

### 3.1. Current HIV Mortality

Table 1 and Figure 2 show mortality data for the most recent calendar year. South Africa had the highest ASDRs for both males (467.7/100,000) and females (391.1/100,000), whereas Egypt had the lowest ASDR for males (0.2/100,000) and Japan for females (0.01/100,000).

In the region of the Americas, the USA had the highest ASDR in 2007 for both males (47.04/100,000) and females (19.02/100,000), whereas Canada had the lowest ASDR in 2005 for both males (18.93/100,000) and females (5.38/100,000).

In the European region, Latvia had the highest ASDR in 2015 for males (57.91/100,000). For females, this was Estonia in 2016 (27.40/100,000), whereas Slovakia had the lowest ASDR in 2014 for males (0.29/100,000), followed by Hungary in 2017 (0.51/100,000). Turkey had the lowest ASDR in 2016 for females (0.24/100,000), followed by the Czech Republic in 2017 (0.31/100,000) and Finland in 2017 (0.41/100,000). Sufficient data were not available for women from Cyprus, Slovenia, Slovakia, and Malta. Out of the 34 European countries, half (17/34) had an ASDR of more than 5.26/100,000. Nine countries had an ASDR of more than 8.8/100,000 in men at the beginning of their respective study periods, ranging from 2001 to 2009, including Austria, France, Italy, Malta, Netherland, Romania, Spain, Serbia, and Switzerland. For the end of the study period, in years ranging from 2013 to 2018, 17 out of the 34 countries had an ASDR value of more than 3.4/100,000, and nine countries (26%) had an ASDR greater than 12/100,000. For females, we had data from 30 countries showing that half (15/30) had an ASDR greater than 1.7/100,000, and around a quarter (8/30) had an ASDR greater than 3/100,000, including Belgium, France, Italy, Luxembourg, Romania, Spain, Switzerland, and Serbia for years between 2001 and 2009 at the beginning of the study periods. For the end of the study periods, at years ranging from 2013 to 2018, half of the 30 countries had an ASDR value of more than 1.3/100,000, and eight countries (27%) had an ASDR with a value of more than 3.5/100,000, including Armenia, Estonia, Latvia, Romania, Georgia, Kyrgyzstan, Republic of Moldova, and Lithuania.

In the Western Pacific region, Malaysia had the highest ASDR in 2014 for males (24.54/100,000) as well as for females (3.79/100,000), followed by Singapore with an ASDR of 11.49/100,000 for males and 1.03/100,000 for females in the year 2017. In comparison, Japan had the lowest ASDR in 2017 for males (0.34/100,000) and females (0.01/100,000). Australia and New Zealand also had lower ASDRs.

In the Eastern Mediterranean region, Bahrain had the highest ASDR in 2014 for males (11.12/100,000) and females (1.6/100,000). In contrast, Egypt had the lowest mortality in 2015 for males (0.15/100,000) and females (0.12/100,000). Jordan and Kuwait showed an ASDR of 1.52/100,000 and 0.48/100,000 for males in 2015 and 2016, respectively. We did not have data for females in the other Eastern Mediterranean countries.

In Africa, out of the two countries included, South Africa had a higher ASDR in 2015 for males (467.27/100,000) as well as for females (391.09/100,000), whereas it was lower for Mauritius in 2018 for males (134.84/100,000) and females (31.22/100,000).

In the Southeast Asia region, Thailand showed an ASDR of 82.44/100,000 and 41.58/100,000 for males and females, respectively, in 2017, which was high compared to many other included countries.

### 3.2. Changes in HIV Mortality between the Start and End of the Study Period

Figure 1, Figure 2 and Figure 3 and Table 1 show HIV mortality at the beginning and the end of the study period. We have reported percentage change (PC) and absolute change (AC) in ASDR. A total of 31/48 countries (64.6%) showed a decrease in male mortality, whereas 35/42 countries (83.3%) showed a decrease in female mortality. Among all 48 countries, Kyrgyzstan had the highest positive PC in male mortality (+6998.6% between 2002 and 2016), while South Africa had the highest positive AC for males (+168,96 between 2007–2015). Spain had the highest PC reduction amongst males (−81.89% between 2001 and 2017), while Thailand had the highest reduction in AC (−192.07 between 2002 and 2017). For female PC, Estonia had the highest increase (+5877.6% between 2001 and 2016), and Australia had the highest reduction (−93.80% between 2001 and 2017). For female AC, South Africa had the highest increase (+78.88 between 2007 and 2015), and Thailand had the highest decrease (−118.28 between 2002 and 2017).

Region-wise, in the Americas, positive PCs were observed in Canadian females (+47.81%) between 2001 and 2005, with a positive AC (1.74). Whereas males had a reduction (−5.31%) in PC and (−1.06) AC. In the USA, males had −29.78% PC and −19.95 of AC, whereas females had −16.82% PC and −3.85 of AC between 2001 and 2007.

In Europe, the maximum positive PC was observed in Kyrgyzstan for males (+6998.63% between 2002 and 2016), while Latvia had the highest positive AC for males with a value of 52.69 (between 2001 and 2015). For females, Estonia had the highest positive values for both PC and AC (+5877.57% and +26.95, respectively between 2001 and 2016). The maximum negative PC and AC values were observed in Spain for males (−81.89% and −46.29, respectively between 2001 and 2017). In females, the highest PC was found to be in Serbia (−85.01% between 2001 and 2017), whereas that of AC was in Romania (−15.71 between 2001 and 2017). For males, 21 (61.8%) countries showed decreased PC and AC in ASDR during our study period. Armenia, Croatia, Czech Republic, Estonia, Finland, Latvia, Malta, Turkey, Georgia, Kyrgyzstan, Lithuania, Republic of Moldova, and Cyprus increased in PC and AC in ASDR in males. Out of 30 European countries with available data for females, 19 (63.3%) showed decreasing PC and AC. In contrast, positive PCs and ACs were observed in Armenia, Croatia, Czech Republic, Estonia, Hungary, Latvia, Turkey, Georgia, Kyrgyzstan, Lithuania, and the Republic of Moldova.

In the Western Pacific region, the overall mortality percentage changes and the absolute changes in ASDR in males were negative in all countries. The largest PC was observed in New Zealand (−70.28% between 2001 and 2017) and the smallest in Malaysia (−7.97% between 2007 and 2014). For AC, the largest negative change was seen in Australia (−6.49 between 2001 and 2017). In females, the highest positive PC and AC values were in Malaysia (+156.71% and +2.31, respectively from 2007 to 2014), and the highest negative PC and AC values were in Australia (−93.80% and −1.04, respectively between 2001 and 2017).

In the Eastern Mediterranean region, Jordan and Kuwait showed positive PC and AC for males. Egypt showed positive PC and AC for females. The maximum positive PC and AC values were found in Jordan for males (+114.48% and +0.81, respectively between 2008 and 2015) and in Egypt for females (+114.38% and +0.06 between 2001 and 2015). While the maximum negative PC was found in Egypt for males (−63.62% between 2001 and 2015), the maximum AC for males was found in Bahrain (−11.10) between 2002 and 2014). For females, Bahrain had negative PC and AC (−74.24% and −4.62, respectively between 2002 and 2014).

In the African region, positive PC was observed in Mauritius (2005–2018) and South Africa (2007–2015) for both genders. Mauritius was found to have a higher PC for males (+436.03%) and females (+587.09%) between 2005 and 2018. ASDR is still increasing in both African countries. AC was also positive in both countries and genders. However, the highest positive AC was in South Africa for both males (+168.96) and females (+78.88) between 2007 and 2015.

Thailand showed decreasing PC and AC in males (−69.97% and −192.07, respectively) and females (−73.98% and −118.28, respectively) between 2002 and 2017.

## 4. Discussion

We analyzed mortality due to HIV/AIDS in 48 countries for males and 42 for females between 2001 and 2018 data from the WHO Mortality Database. At the end of the study period, the five leading countries with the highest ASDRs in males were South Africa, Mauritius, Thailand, Latvia, and Estonia. The five countries with the highest ASDRs in females were South Africa, Thailand, Mauritius, Estonia, and Latvia. Egypt, Japan, Kuwait, Hungary, and Sweden had the lowest ASDR in males. In females, Japan, Australia, Egypt, Turkey, and the Czech Republic had the lowest ASDRs. In males, 17 countries had percentage increases in ASDRs over the study period, with the highest increases in Kyrgyzstan, Georgia, Latvia, the Republic of Moldova, and Estonia (Range: +6998.62%–+436.49%). The same countries had an absolute increase in ASDR, with South Africa and Mauritius being the top two (+168.95 and +109.68, respectively). However, some of the countries with massively large percentage changes (ex Kyrgyzstan: +6998.62%) had absolute changes that were comparable to other countries (AC of +29.29). This seems to be likely owing to the exceedingly small proportion of HIV-related deaths during the beginning year (e.g., Kyrgyzstan ASDR: 0.42/100,000 in 2002). In females, 13 countries had a percentage change and absolute change increase, with the highest PC increase in Kyrgyzstan, Georgia, Latvia, Republic of Moldova, and Estonia (Range: +6998.62%–+676.97%) and the highest AC increase in the same countries as well as South Africa and Mauritius. This multifold mortality increase in some countries is worrisome. This indicates a significant risk that the progress made in slowing the HIV epidemic could be reversed without continued efforts.

Thirty-one countries had both percentage change and absolute change decreases in males. Twenty-five countries had percentage change and absolute change decreases in females, which is encouraging. In 2018, the disparity in ASDR rates in males, measured using rate ratios, between the lowest rates observed in Egypt and highest rates observed in South Africa was 3034. In females, rate ratios between the lowest rates observed in Japan and highest rates observed in South Africa were 41,211. These large differences help demonstrate that countries are on remarkably diverse trajectories with regard to the burden of HIV/AIDS. Trends from the Global Burden of Disease (GBD) database have shown high mortality in Sub-Saharan Africa and South Asia, as well as increased mortality in South East Asia. Apart from these regions, in our study, we have also observed increasing ASDRs in Eastern Europe and the Central Asia region (Latvia, Lithuania, Estonia, Kyrgyzstan, Georgia, and Armenia) as well as in the Central Europe region (the Czech Republic and Croatia) [18]. This finding highlights the importance of the harmonization of ‘real world’ data and the increasing need for independent multinational registries to better understand health-system-level differences in policy, clinical practice, and outcomes.

Seventy-six million people have tested positive for HIV since the start of the epidemic. Thirty-eight million people globally were living with HIV in 2019, of whom 81% knew their status, 67% were accessing treatment, and 59% were virally suppressed. From 2010 to 2019, new HIV infections declined by 23%, from 2.1 million to 1.7 million. The global community has made considerable progress in addressing this unprecedented epidemic [1]. In our study, high HIV mortality areas were concentrated in resource-limited countries such as the African (South Africa, Mauritius) and the South East Asian regions (Malaysia). The WHO Mortality Database does not have data of prevalence or incidence. Studies in the past using the GBD database have shown that in 2015, 75% of new cases were in Sub-Saharan Africa, followed by South Asia (8.5%) and Southeast Asia (4.7%). The overall high prevalence and incidence could be affecting this high mortality substantially [18]. Antiretroviral therapy (ART) has extended the life expectancy for most people living with HIV. The percentage of people receiving ART has changed over time, overall and by region. A Study by Lau et al. showed that in 2014–2015, 22% of HIV-positive patients in Europe did not receive ART [19]. In low- and middle-income countries, two-thirds of HIV/AIDS-related deaths occur in individuals not on ART [20]^20^. The number of individuals who lost their lives to HIV/AIDS in 2015 is 45% lower than the peak of the epidemic in 1998 [13,21]. HIV/AIDS remains a significant cause of death despite these gains and global trends mask persistent regional and subregional variation [22].

As of June 2020, access to ART for HIV-positive patients was 72% in South Africa, 82% in Mauritius, and 80% in Thailand, compared to the global average of 67% [21]. Out of the 48 countries studied, ASDRs were highest in these countries, with 467.27/100,000, 134.84/100,000, and 82.44/100,000 in males, respectively, and 391.10/100,000, 41.58/100,000, and 31.22/100,000 in females, respectively [12,23]. The disease continues to claim lives in regions with high treatment coverage levels, this suppors a complex nature of the care model. After more than a decade of ART scale-up and more than 19.5 million people on treatment, an increase in drug resistance is inevitable. Data from several national surveys from low-income and middle-income countries have shown an increased prevalence of pretreatment HIV drug resistance to non-nucleoside reverse-transcriptase inhibitors. Drug resistance testing is largely not available in low- and middle-income countries and might be contributing to the high mortality despite good ART coverage [24]. Non-HIV-related conditions are also emerging as prominent health concerns in settings where ART is widely available. A prospective cohort study of PLHIV patients in Spain found that the most common non-AIDS events developing in these patients were psychiatric, hepatic, renal, cardiovascular, and malignant diseases [25]. Interestingly, the leading causes of non-AIDS-associated mortality in HIV-positive individuals were malignancy, most commonly: lung, followed by liver, with which hepatitis C virus was frequently found associated [21]. Additionally, hepatitis B virus coinfections are high in endemic regions (East and Southeast Asia) [26]. A large body of evidence indicates that HIV-positive individuals are at above-average risk for cardiovascular disease [27,28]. Coinfection with tuberculosis remains a major public health concern and is the main cause of high mortality in HIV globally, particularly in resource-limited settings [29]. Older age, low CD4 count, and cigarette use are independent predictors of mortality in HIV-positive individuals [30]. Additional risk factors for non-AIDS events include a late stage at initial treatment, coinfection with hepatitis C virus, and injecting drug abuse [31]. This situation raises the question of how the health needs of HIV patients should be addressed beyond access to ART. Given these trends, the management of HIV patients in low- and middle-income countries should focus on reasons of virologic failure in patients receiving ART, providing universal access to better quality health services, and timely identification and management of potential non-AIDS complications.

The rapid expansion of ART began in 2004 [32]. One of the areas most influenced by the implementation and availability of ART was Eastern and Southern Africa. HIV and population mortality rates saw a steep decline [22]. Initially, HIV-positive individuals would only qualify for ART if they were WHO stage 3 or 4 or had a CD4 count less than 200 [26]. These requirements have been evolving ever since. A recent study in Zimbabwe demonstrated the effectiveness of “Test and Treat” protocol; of 972 people newly diagnosed HIV-positive, 94% were enrolled in HIV care, 79% were initiated on ART, and 71% were retained alive on ART at three months [33]. In 2014, UNAIDS also created the “90–90-90” approach in a further attempt to eradicate the virus [34]. Treatment, as well as pre-exposure prophylaxis (PrEP), has seen constant advancements influencing the mortality trends [29]. The Centers for Disease Control and Prevention recommends high-risk individuals take PrEP to limit their chances of contracting disease [35]. ART initiation is also one of the most effective ways to prevent and help resolve opportunistic infections. At the time of diagnosis of HIV and before initiation of ART, all individuals must be screened for latent tuberculosis (TB) infection [36], given the potential for reactivation of latent TB and the need for a mycobacterial treatment regimen before the initiation of ART [36].

Region-specific increases in HIV mortality are well known. In our data, South Africa, Thailand, and Mauritius were all found to have increased ASDRs. Several explanations for this pattern may exist. Despite the expansion and the availability of ART, resource-limited areas such as those in South Africa have associated delays in diagnosis and initiation of treatment, which ultimately led to increased mortality [17]. In addition, high-risk activities such as increased IVDA and sex trafficking are more common in certain areas of Asia and Africa and likely contribute to the increased HIV prevalence and mortality [37]. Particular areas where TB is more prevalent have an increased rate of reactivation which increases mortality risk in the HIV-positive population [36].

The strengths of this investigation include the use of annual mortality data collected from national surveillance statistics from the WHO. These data have made it possible to assess population-level trends over an extended observation period, allowing comparisons in directions rather than total annual mortality rates. Despite this study’s strengths, there are several limitations which should be considered when interpreting the results. Out of 194 countries, data of only 48 countries have been analyzed due to the non-availability of data. Additionally, the start and end years of countries varied based on the data availability. In 2015, 75.4% of new cases were in Sub-Saharan Africa [18]. We understand that South Africa and Mauritius do not represent the population of Sub-Saharan Africa. However, due to the lack of availability and completeness of the data, we could not include these countries in our study. We did not attempt to assess the prevalence of morbidity associated with HIV, as the WHO Mortality Database does not report this. ICD coding for HIV mortality presents many challenges and is one of the limitations of all database studies. To achieve comparability over time, the WHO has tried to standardize ICD-10 coding for HIV with detailed guidelines. However, issues still exist when medical certification is not possible, especially in Sub-Saharan Africa, as they occur at home without a medical doctor. The WHO has developed computerized coding of verbal autopsy to address this issue using algorithmic methods. Validation studies showed that based on the symptoms indicative of AIDS-defining illnesses from verbal autopsy, 89% of deaths among HIV-positive individuals are attributable to HIV [38]. Finally, as with any observational study, causal statements regarding the observed trends cannot be made. However, one notable strength of using longitudinal data is the ability to comment on overall trends within individual countries after standardization and to report these differences between health systems. The discussion is provided to assist future researchers, policymakers, and public health experts in focusing their efforts.

## 5. Conclusions

Although there has been a decrease in mortality attributed to HIV among most of the countries studied, a rising trend remains in a number of developing countries. A renewed and heightened commitment to address this ongoing epidemic by these countries, healthcare agencies, and the global community is called for.

## Figures and Tables

**Figure 1 tropicalmed-06-00173-f001:**
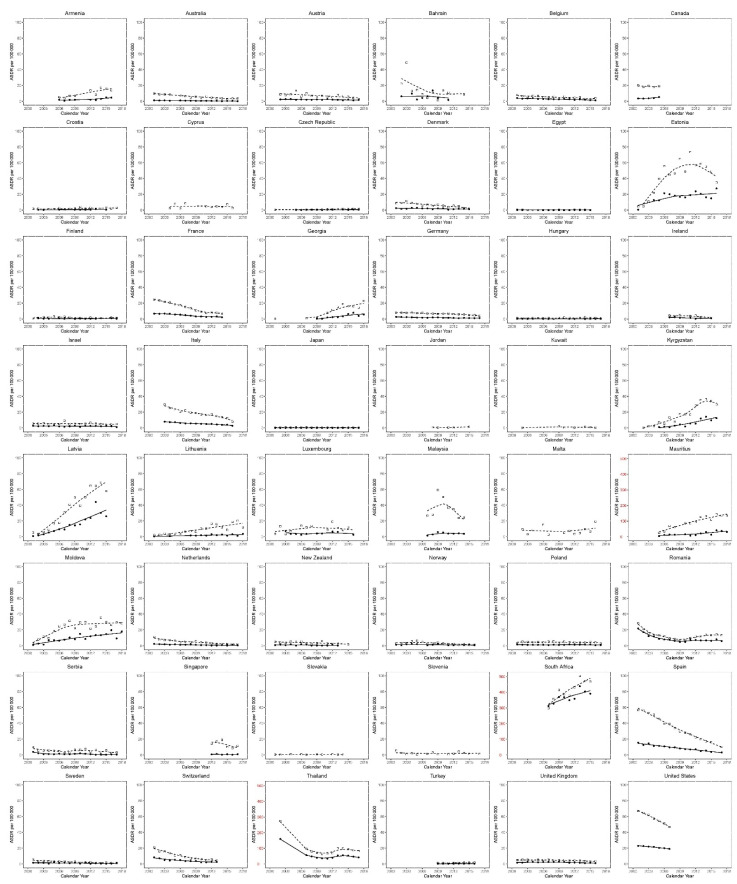
Trends in age-standardized death certification rates per 100,000 for HIV. Squares indicate male mortality, whereas circles indicate females.

**Figure 2 tropicalmed-06-00173-f002:**
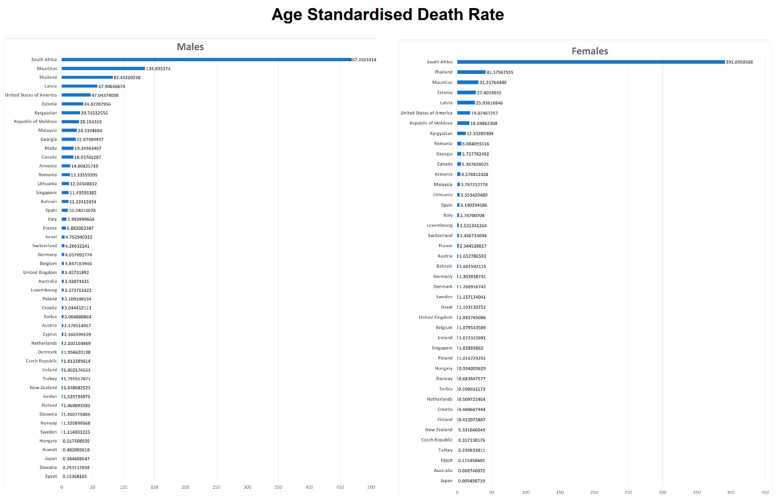
Age-standardized death rate in males and females, 2018.

**Figure 3 tropicalmed-06-00173-f003:**
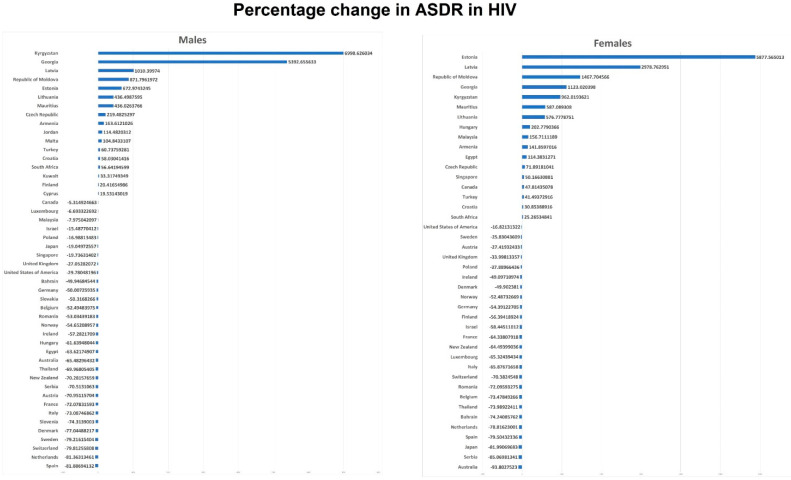
Percentage changes in age-standardized death rate in males and females.

**Table 1 tropicalmed-06-00173-t001:** Age-standardized death rate in males and females.

Region	Country	Start Point				End Point				Overall Percentage Change in Mortality		Absolute Changes in ASDR	
		Year of Diagnosis	Age Adjusted Rate in Males	Year of Diagnosis	Age Adjusted Rate in Females	Year of Diagnosis	Age Adjusted Rate in Males	Year of Diagnosis	Age Adjusted Rate in Females	Males	Females	Males	Females
Americas	Canada	2001	20.00063136	2001	3.644861272	2005	18.93761287	2005	5.387628025	−5.3149247	47.8143508	−1.0630185	1.74276675
	United States of America	2001	66.99524774	2001	22.87205209	2007	47.04374008	2007	19.02467257	−29.780482	−16.821313	−19.951508	−3.8473795
Europe	Israel	2001	5.623963084	2001	2.654634041	2017	4.752940322	2017	1.103130252	−15.487704	−58.44511	−0.8710228	−1.5515038
	Armenia	2006	5.31396588	2006	1.893003422	2016	14.00825719	2016	4.578412428	163.612103	141.859702	8.69429131	2.68540901
	Austria	2002	8.876480762	2002	2.277171682	2017	2.578514957	2017	1.652786593	−70.951157	−27.419324	−6.2979658	−0.6243851
	Belgium	2001	8.098412731	2001	4.070445526	2016	3.847163946	2016	1.079543509	−52.49484	−73.478493	−4.2512488	−2.990902
	Croatia	2001	1.926497586	2001	0.367285945	2017	3.044452113	2015	0.480607944	58.0304142	30.8538892	1.11795453	0.113322
	Czech Republic	2001	0.567257814	2001	0.184498712	2017	1.812289614	2017	0.317138176	219.48253	71.8918104	1.2450318	0.13263946
	Denmark	2001	8.52367743	2001	2.532888324	2015	1.956620198	2015	1.268916742	−77.044882	−49.902381	−6.5670572	−1.2639716
	Estonia	2001	4.505049968	2001	0.458469351	2016	34.82287956	2016	27.4053035	672.974324	5877.56501	30.3178296	26.9468342
	Finland	2001	1.213200001	2001	0.945002237	2017	1.460893585	2017	0.412075887	20.4165499	−56.394189	0.24769358	−0.5329263
	France	2001	24.64773747	2001	6.574320069	2014	6.882063387	2014	2.344528817	−72.078316	−64.338079	−17.765674	−4.2297913
	Germany	2001	8.115361794	2001	2.858964748	2017	4.057091774	2017	1.303938741	−50.007259	−54.391227	−4.05827	−1.555026
	Hungary	2001	1.349066553	2001	0.328295393	2017	0.517508939	2017	0.994009629	−61.63948	202.779037	−0.8315576	0.66571424
	Ireland	2007	4.218787777	2007	2.106605118	2015	1.802174553	2015	1.072322891	−57.282171	−49.09711	−2.4166132	−1.0342822
	Italy	2003	29.70363342	2003	8.050242431	2016	7.993999664	2016	2.74700704	−73.087469	−65.876717	−21.709634	−5.3032354
	Latvia	2001	5.214929781	2001	0.842421741	2015	57.90656674	2015	25.93616846	1010.39974	2978.76295	52.691637	25.0937467
	Luxembourg	2001	3.615713815	2001	7.300063736	2016	3.373702422	2016	2.531341314	−6.6933227	−65.324394	−0.2420114	−4.7687224
	Malta	2002	9.446066172	NA	NA	2016	19.34963467	NA	NA	104.843311	NA	9.9035685	NA
	Netherlands	2001	11.27960536	2001	2.406207509	2017	2.102164869	2017	0.509725464	−81.363135	−78.81623	−9.1774405	−1.896482
	Norway	2001	2.912812422	2001	1.439295097	2016	1.320899568	2016	0.683847577	−54.65209	−52.487327	−1.5919129	−0.7554475
	Poland	2001	3.74537848	2001	1.636972721	2017	3.109108534	2017	1.016729251	−16.988135	−37.889664	−0.6362699	−0.6202435
	Romania	2001	28.39437744	2001	21.80360756	2017	13.33559205	2017	6.084093316	−53.034392	−72.095933	−15.058785	−15.719514
	Slovakia	2001	0.589962431	NA	NA	2014	0.293112058	NA	NA	−50.316827	NA	−0.2968504	NA
	Slovenia	2001	5.453439411	NA	NA	2017	1.400775885	NA	NA	−74.3139	NA	−4.0526635	NA
	Spain	2001	56.53435158	2001	15.5656934	2017	10.24010028	2017	3.190294186	−81.886941	−79.504323	−46.294251	−12.375399
	Sweden	2001	5.359937843	2001	1.560119784	2017	1.114001225	2017	1.157134041	−79.216154	−25.830436	−4.2459366	−0.4029857
	Switzerland	2001	21.13354147	2001	8.328558892	2013	4.26632141	2013	2.466714694	−79.812558	−70.382455	−16.86722	−5.8618442
	United Kingdom	2001	4.73948267	2001	1.640506768	2016	3.45731892	2016	1.082765086	−27.052821	−33.998134	−1.2821637	−0.5577417
	Turkey	2009	1.114560098	2009	0.169501371	2016	1.791517071	2016	0.239833811	60.7375928	41.4937292	0.67695697	0.07033244
	Georgia	2001	0.4164297	2001	0.470783848	2018	22.87304937	2018	5.757782492	5392.65563	1123.0204	22.4566197	5.28699864
	Kyrgyzstan	2002	0.418606719	2002	1.161189662	2016	29.71532556	2016	12.33205904	6998.62603	962.019362	29.2967188	11.1708694
	Lithuania	2001	2.244815687	2001	0.525051072	2018	12.04340832	2018	3.553429489	436.498759	576.777875	9.79859263	3.02837842
	Republic of Moldova	2001	2.89714192	2001	1.151277063	2018	28.154315	2018	18.04862308	871.796197	1467.70457	25.2571731	16.897346
	Serbia	2001	10.1905643	2001	3.957969109	2017	3.004880864	2017	0.590932173	−70.513106	−85.069813	−7.1856834	−3.3670369
	Cyprus	2004	2.147216974	NA	NA	2016	2.566599159	NA	NA	19.5314302	NA	0.41938218	NA
Western Pacific	Japan	2001	0.425704112	2001	0.052743399	2017	0.344608647	2017	0.009498719	−19.049726	−81.990697	−0.0810955	−0.0432447
	Malaysia	2007	26.6661012	2007	1.475281941	2014	24.5394684	2014	3.787212778	−7.9750421	156.711119	−2.1266328	2.31193084
	Singapore	2012	14.31788695	2012	0.685039559	2017	11.49206382	2017	1.02869862	−19.736314	50.1663088	−2.8258231	0.34365906
	Australia	2001	9.910307425	2001	1.109300056	2017	3.42074435	2017	0.068746072	−65.482964	−93.802752	−6.4895631	−1.040554
	New Zealand	2001	5.54767863	2001	0.934112429	2015	1.648682625	2015	0.331666049	−70.281577	−64.49399	−3.898996	−0.6024464
Eastern Mediterranean	Egypt	2001	0.422453104	2001	0.054039048	2015	0.15368105	2015	0.115850601	−63.621749	114.383127	−0.2687721	0.06181155
	Bahrain	2002	22.22462627	2002	6.217567695	2014	11.12412654	2014	1.601592115	−49.946845	−74.240858	−11.1005	−4.6159756
	Jordan	2008	0.711357947	NA	NA	2015	1.525734975	NA	NA	114.482031	NA	0.81437703	NA
	Kuwait	2002	0.361547165	NA	NA	2016	0.482005618	NA	NA	33.3174935	NA	0.12045845	NA
South East Asia	Thailand	2002	274.5146875	2002	159.8555757	2017	82.44210258	2017	41.57967555	−69.968054	−73.989224	−192.07258	−118.2759
Africa	Mauritius	2005	25.15461549	2005	4.543462418	2018	134.835374	2018	31.21764449	436.026377	587.089308	109.680758	26.6741821
	South Africa	2007	298.3021747	2007	312.2132831	2015	467.2663314	2015	391.0950568	56.641946	25.2653484	168.964157	78.8817738

## Data Availability

We extracted HIV mortality data from the WHO Mortality Database from 2001 to 2018 for the member nations with available data.

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
