# Peer review of "Trends of HIV Mortality between 2001 and 2018: An Observational Analysis"

_tropicalmed, 2021, doi:10.3390/tropicalmed6040173_

Round 1

Reviewer 1 Report

In their manuscript, Jani et al describe mortality rates across 48 countries with complete data in the WHO Mortality Database across periods ranging from 2001-2018.

One of the major issues with studying mortality outcomes of chronic infections in national or worldwide public databases is that without knowing the incidence or yearly prevalence of the infection, the mortality rates are fairly meaningless to the population for which the estimate is useful. Here, the HIV prevalence varies substantially across countries, and it is no surprise that some countries have a higher burden of disease in the general population simply because the prevalence is higher. The discussion on this point is very limited.

Cross country comparisons are then problematic. The overall specificity of mortality to HIV-positive individuals is a real limitation, which makes other efforts, such as the Global Burden of Diseases study (GBD 2015 HIV Collaborators, Lancet HIV, 2016) or the WHO’s Global Health Observatory, much more informative. These results should be compared to these studies.

Another issue, at least at the country level, is that identifying HIV as the cause of death depends on whether the individual had been tested for HIV. There are likely differences in HIV testing rates across settings, hence there could be some misclassification of death. This potential for misclassification needs to be discussed. The authors never really go into detail on how the endpoint was adjudicated by the WHO. As the authors mentioned in the discussion, there are indeed differences in ART coverage between countries, yet these figures are never really relayed back to the endpoint of mortality.

There are several time-series models available that could be fit to give an effect estimate of the impact of the proportion undergoing ART, but this would require that the authors retrieve data from other sources. The description of the statistical methods used is wanting. It seems that LOWESS was only used to make plots and not to model changes across countries within the same region. “Visual inspection” is unclear. “Software requires continuous data throughout the observation period” can be remedied with a variety of methods. “…to avoid excess imputation” is odd to bring up when imputation was never discussed. How was “percent change” calculated? Would it not be more appropriate to use absolute change in mortality and not relative %, as higher % might be an artifact of very low rates of the baseline comparator? I would recommend that the authors read how others have used the WHO Mortality Database (particularly with respect to the LOESS functions: Ebmeier S et al, Lancet, 2017; Barco S et al, Lancet Res Med, 2020).

Finally, it is striking that sub-Saharan Africa, having the highest HIV prevalence and burden of HIV in terms of mortality worldwide, is only represented by South Africa and Mauritius. Would the same results be observed in western or eastern Africa?

Minor comments:

- ln 67. Methods. “completeness of data” is defined how?

- ln 78-79. The definition for estimated level of coverage is confusing. When was it reported in the Results?

- The “percent change in ASDR” concerns different time periods. Estimates need to be given in average yearly rates of change, or another method should be entertained.

- ln 267. “Injecting” should be used instead of “intravenous”.

- ln 269-70. Hepatitis B virus infection is probably more likely to cause liver-related death than HCV in sub-Saharan Africa and Southeast Asia.

Author Response

Reviewer-1

In their manuscript, Jani et al describe mortality rates across 48 countries with complete data in the WHO Mortality Database across periods ranging from 2001-2018.

One of the major issues with studying mortality outcomes of chronic infections in national or worldwide public databases is that without knowing the incidence or yearly prevalence of the infection, the mortality rates are fairly meaningless to the population for which the estimate is useful. Here, the HIV prevalence varies substantially across countries, and it is no surprise that some countries have a higher burden of disease in the general population simply because the prevalence is higher. The discussion on this point is very limited. Cross country comparisons are then problematic. The overall specificity of mortality to HIV-positive individuals is a real limitation, which makes other efforts, such as the Global Burden of Diseases study (GBD 2015 HIV Collaborators, Lancet HIV, 2016) or the WHO’s Global Health Observatory, much more informative. These results should be compared to these studies.

Epidemiological transition has brought reduction in the infectious diseases in many developed countries. However, diseases like AIDS, tuberculosis and influenza have emerged and re-emerged. Even though, incidence and prevalence are not available through WHO mortality database, it reports an actual data of mortality with no estimation which is the case in Global Burden of Disease database. Both the data-bases have their own benefits and limitations. With actual reporting, we have the real-world data although there is always a possibility of under-reporting. Estimation through GBD would help us in filling this gap. However, it is always an estimation and not the real-world data. We feel that the literature from different databases can help us in understanding these epidemiological trends in more detailed manner. We highly appreciate the suggestion of the reviewer for the comparison to previous studies and therefore, we have added few comparison in our discussion now. 

“Trends from the Global Burden of Disease (GBD) database have still shown high mortality in Sub-Saharan Africa and South Asia, as well as increasing mortality in South-East Asia. Apart from these regions, in our study we have also observed increasing ASDRs in Eastern Europe and central Asia region (Latvia, Lithuania, Estonia, Kyrgyztan, Georgia and, Armenia) as well as in Central Europe region (Czech Republic and Croatia) 18. This finding highlights the importance of harmonization of ‘real world’ data and the increasing need for independent multinational registries, in order to better understand health-system level differences in policy, clinical practice, and outcomes.”

Regarding effect of prevalence, we have added following sentences:

“The WHO Mortality Database does not have data pertaining to prevalence or incidence. Studies in the past using GBD database have shown that, in 2015, 75% of new cases were in Sub-Saharan Africa followed by South Asia (8.5%) and Southeast Asia (4.7%). The overall high prevalence and incidence can be affecting this high mortality substantially 18.”

Another issue, at least at the country level, is that identifying HIV as the cause of death depends on whether the individual had been tested for HIV. There are likely differences in HIV testing rates across settings, hence there could be some misclassification of death. This potential for misclassification needs to be discussed. The authors never really go into detail on how the endpoint was adjudicated by the WHO. As the authors mentioned in the discussion, there are indeed differences in ART coverage between countries, yet these figures are never really relayed back to the endpoint of mortality.

Following sentences added to the manuscript:

“ICD coding for HIV mortality presents a number of challenges. In order to achieve comparability over time, WHO has designed a standardized verbal autopsy instrument that can serve different purposes. Although, due to the current usage of database, this is one of the limitations 38.”

There are several time-series models available that could be fit to give an effect estimate of the impact of the proportion undergoing ART, but this would require that the authors retrieve data from other sources. The description of the statistical methods used is wanting. It seems that LOWESS was only used to make plots and not to model changes across countries within the same region. “Visual inspection” is unclear. “Software requires continuous data throughout the observation period” can be remedied with a variety of methods. “…to avoid excess imputation” is odd to bring up when imputation was never discussed. How was “percent change” calculated? Would it not be more appropriate to use absolute change in mortality and not relative %, as higher % might be an artifact of very low rates of the baseline comparator? I would recommend that the authors read how others have used the WHO Mortality Database (particularly with respect to the LOESS functions: Ebmeier S et al, Lancet, 2017; Barco S et al, Lancet Res Med, 2020).

We have edited the methodology section and removed sentences that were not directly applicable to our statistics. We did not perform any imputation. For percentage change, we calculated as [(End ASDR – Start ASDR)/Start ASDR] *100 for each gender and country. We have also added absolute number in the table for the comparison with the relative change.

The updated methods paragraph is copied below:

“We computed male and female mortality rates and used a locally weighted scatter-plot smoothing (LOWESS) plot fit to the rates of male and female mortality using SAS v9.4 (SAS, Cary, NC) (Figure 1). LOWESS was only used to make plots for visualization of country-specific trends and not to model changes across the countries. Mortality data were missing in a small subset of countries in the WHO Mortality Database for one or more calendar years. If a country had more than three consecutive years of data missing during the observation period, it was excluded from the analysis. There were no other modifications to the data. Changes in ASDRs over the observation period are calculated as crude absolute differences between first and last data points for the earliest and most recent years available. Like our previous studies, percentage change was calculated as [(End ASDR – Start ASDR)/Start ASDR]*100 for each gender and country 16,17. We have also provided absolute changes in the ASDR. This can serve as a comparator to the percentage changes especially for the countries with low baseline mortality.”

Finally, it is striking that sub-Saharan Africa, having the highest HIV prevalence and burden of HIV in terms of mortality worldwide, is only represented by South Africa and Mauritius. Would the same results be observed in western or eastern Africa?

We understand that South Africa and Mauritius are not entirely representative of the population of Sub-Saharan Africa. However, due to the availability and completeness of the data reported by WHO, the other countries did not meet the data quality and completeness criteria for inclusion in the study. Without analysis it would be difficult to comment if other regions in Africa have similar results. We have included this as a limitation in the manuscript.

Minor comments:

- ln 67. Methods. “Completeness of data” is defined how?

Further details about completeness of data is now added to the methods sections:

“The proportion of all deaths which are registered in the population covered by the vital registration system for a country (referred to as completeness) has been estimated by WHO for the latest available year. For Member States with incomplete vital registration systems, demographic techniques have been used by WHO to estimate the level of completeness of death recording for the specified population to allow the calculation of death rates.”

- ln 78-79. The definition for estimated level of coverage is confusing. When was it reported in the Results?

Further details about level of coverage is now added to the methods sections:

“The estimated level of coverage for deaths with a recorded cause for death is calculated by dividing the total deaths reported for a country-year from the vital registration system by the total estimated deaths for that year for the national population. The national population estimates used are those of the UN Population Division. Best estimates of death rates by age and sex, adjusted for incompleteness and incomplete coverage), are applied to the national population data to obtain total estimated deaths. WHO estimated coverage for a Member State may be less than 100% due to incompleteness of registration, or to coverage of only some parts of the national population, or to differences between the vital registration population and the UN estimated de-facto population.”

- The “percent change in ASDR” concerns different time periods. Estimates need to be given in average yearly rates of change, or another method should be entertained.

We calculated percentage change by [(End ASDR – Start ASDR)/Start ASDR]*100 for each gender and country. This equation is described in the methods section of the manuscript.

- ln 267. “Injecting” should be used instead of “intravenous”.

We have changed the word to “injecting”.

- ln 269-70. Hepatitis B virus infection is probably more likely to cause liver-related death than HCV in sub-Saharan Africa and Southeast Asia.

Following sentence added to the manuscript:

“Also, Hepatitis B virus co-infections are high in endemic regions (East and South-east Asia) or through medical or traditional scarification procedures in Africa26.”

Reviewer 2 Report

Introduction:
Rationale / Justification
A paragraph needs to be added, why this study is important?

Objectives:
To describe current mortality rates for HIV is a broader statement, it will be better to divide objectives of the study into General objectives and specific objectives.

Methods:
When this study was conducted?

Discussion:
Countries with high treatment coverage have higher mortality rates, reasons need to be properly explained.

Weaknesses of study
Out of total194 countries, data of only 48 countries have been analyzed due to the non-availability of data.

Author Response

Reviewer-2

Introduction:
Rationale / Justification

A paragraph needs to be added, why this study is important?

Thank you for this suggestion. We have added the following paragraph to the manuscript relating to the study’s importance.

“In year 2014, the Joint United Nations Program on HIV and AIDS (UNAIDS) and partners set the ‘90-90-90 targets’; aiming to diagnose 90% of all HIV people, provide antiretroviral therapy (ART) for 90% of those diagnosed and achieve viral suppression for 90% of those treated, by 2020 12,13. This led to more widespread and earlier administration of ART. It is important to examine how the setting of these targets, and the subsequent efforts to meet them, have impacted global HIV mortality trends.”

Objectives:
To describe current mortality rates for HIV is a broader statement, it will be better to divide objectives of the study into General objectives and specific objectives.

We have divided objectives into primary and secondary. We have also updated it in the manuscript.

“The primary objective of this study was to describe current ASDRs for HIV and the secondary objectives were to compute the absolute and percentage changes in ASDRs categorized by country and gender through an observational analysis of HIV mortality data from 2001 to 2018 from the World Health Organization (WHO) Mortality Database. We also compared the mortality trends from HIV in 48 nations between 2001 and 2018.”

Methods:
When this study was conducted?

The data was extracted in October 2020 with available  data from the WHO Mortality Database from 2001 to 2018.

We have added “October 2020” as the month of data extraction.

Discussion:
Countries with high treatment coverage have higher mortality rates, reasons need to be properly explained.

Thank you for the suggestion. We have expanded on the reasons. One of the reason can be increasing drug resistance and lack of available testing for the same. Second reason can be emerging non-HIV related conditions in People living with HIV including malignancy and other infectious as well as non-communicable diseases.

“After more than a decade of ART scale-up and more than 19.5 million people on treatment, an increase in drug resistance is inevitable. Data from several national surveys from low-income and middle-income countries have shown increase prevalence of pretreatment HIV drug resistance to non-nucleoside reverse-transcriptase inhibitors above 10%. Drug resistance testing is largely not available in low- and middle-income countries and might be contributing to the high mortality despite good ART coverage24 . Non-HIV-related conditions are also emerging as prominent health concerns in settings where ART is widely available. A prospective cohort study of PLHIV patients in Spain found that the most common non-AIDS events developing in these patients were psychiatric, hepatic, renal, cardiovascular, and malignant diseases.25 Interestingly, the leading causes of non-AIDS-associated mortality in HIV-positive individuals were malignancy, most commonly lung, followed by liver disease, with which hepatitis C virus was frequently found associated21. Also, Hepatitis B virus co-infections are high in endemic regions (East and South-east Asia) or through medical or traditional scarification procedures in Africa26. A large body of evidence indicates that HIV-positive individuals are above-average risk for cardiovascular disease 27, 28. Co-infection with tuberculosis remains a major public health concern and is the main cause of high mortality in HIV globally, particularly, in resource-limited settings.34 Older age, low CD4 count, and cigarette use are independent predictors of mortality in HIV-positive individuals 29. Additional risk factors for non-AIDS events include a late stage at initial treatment, coinfection with hepatitis C virus, and injecting drug abuse 30. This situation raises the question of how the health needs of HIV patients should be addressed beyond the access of ART. Given these trends, the management of HIV patients in low- and middle-income countries should focus on reasons of virologic failure in patients receiving ART and provide universal access to the health services of sufficient quality to be effective without any financial hardship on the patients to optimization timely identification and management of potential non-AIDS complications and comorbid conditions.”

Weaknesses of study

Out of total194 countries, data of only 48 countries have been analyzed due to the non-availability of data.

This is based on the inclusion criteria relating to availability and quality of data. A sentence highlighting this has been added to the limitations section of the study.

Round 2

Reviewer 1 Report

I thank the authors for carefully answering my comments. The manuscript is clearer in many aspects, but there are still some issues that need to be reconciled:

- With regards to the comment on HIV prevalence, the added discussion points are sufficient. The estimated prevalence of HIV-positive individuals should be added to table 1 to make this point even clearer.

- The added text “WHO has designed a standardized verbal autopsy instrument that can serve different purposes.” is vague. Could the authors please provide statistics from validation studies? Or something more tangible? Particularly with respect to HIV? If such information is not available, then it needs to be made explicit.

- The statistical analysis section has been very much improved. The only issue is that the begin and start years for calculating the percent change in ASDR are not the same across countries, hence there could be some bias. The authors could simply conduct a sensitivity analysis showing what would happen when using the same start and end year across all counties (or a majority thereof). The limitation must be addressed.

- I am very pleased that absolute change is given in Table 1. However, these figures tell a different story. Some of the countries with massively large percent changes have absolute changes that are very much comparable to other countries (likely owing to the exceedingly small proportion of HIV-related deaths during the begin year). I was surprised to see absolutely no discussion this – and given the data, absolute differences would seem to be more meaningful (Rothman & Greenland, Modern Epidemiology). Please amend your results and discussion accordingly.

Minor comment:

- “through medical or traditional scarification procedures in Africa26” Most HBV infections in SSA are from horizontal transmission during adolescence.

Author Response

Hiv revisions:

Thank the authors for carefully answering my comments. The manuscript is clearer in many aspects, but there are still some issues that need to be reconciled:

  • With regards to the comment on HIV prevalence, the added discussion points are sufficient. The estimated prevalence of HIV-positive individuals should be added to table 1 to make this point even clearer.

o Response:            WHO mortality database does not have the data for HIV prevalence. We had highlighted this in our discussion.

  • The added text “WHO has designed a standardized verbal autopsy instrument that can serve different purposes.” is vague. Could the authors please provide statistics from validation studies? Or something more tangible? Particularly with respect to HIV? If such information is not available, then it needs to be made explicit.
  •  Response: We have tried explaining it with more details. The same reference on “Obtaining cause-of-death information for HIV/AIDS” has validation studies in Box 5. Details are explained as below in the manuscript
  • “ICD coding for HIV mortality presents a number of challenges and it is one of the limitations of all database studies. However, to achieve comparability over time, WHO has tried to standardize ICD-10 coding for HIV with detailed guidelines. Although is-sues still exist when medical certification is not possible especially in sub-Saharan Africa as they occur at home without the presence of medical doctor. WHO has developed computerized coding of verbal autopsy to address this issue using algorithmic methods. Validation studies showed that based on the symptoms indicative of AIDS-defining illnesses from verbal autopsy, 89% of deaths among HIV-positive individuals are attributable to HIV 38.”
  • The statistical analysis section has been very much improved. The only issue is that the begin and start years for calculating the percent change in ASDR are not the same across countries, hence there could be some bias. The authors could simply conduct a sensitivity analysis showing what would happen when using the same start and end year across all counties (or a majority thereof). The limitation must be addressed.

 Response: On request of reviewer, we did analysis of the data keeping 2007 as the start date and 2014 as the end data which would include majority of the countries. We had to exclude USA and Canada as their end dates were before that. However, the data generated would create lots of bias and would not give glimpse of the whole actual data. We have attached the table at the bottom. As this would be rather bias creating results, we have not added it to the manuscript.

However, we have highlighted in our limitation “Also, the start and end year of countries in our study varied based on the availability of the data.”

  • I am very pleased that absolute change is given in Table 1. However, these figures tell a different story. Some of the countries with massively large percent changes have absolute changes that are very much comparable to other countries (likely owing to the exceedingly small proportion of HIV-related deaths during the begin year). I was surprised to see absolutely no discussion this – and given the data, absolute differences would seem to be more meaningful (Rothman & Greenland, Modern Epidemiology). Please amend your results and discussion accordingly.

 Response: Thanks for highlighting this part. We have now extensively added absolute change in results as well as discussion.     

Minor comment:

  • “through medical or traditional scarification procedures in Africa26” Most HBV infections in SSA are from horizontal transmission during adolescence.

 Response: We have made the changes in the manuscript.

“Also, Hepatitis B virus co-infections are high in endemic regions (East and South-east Asia) 26”

Table with 2007 and 2014 as end dates.

Region

Country

Overall percentage change in mortality in males

Absolute changes in males

Overall percentage change in mortality in males

Absolute changes in males

Till original end date

till 2014

Till original end date

till 2014

Till original end date

till 2014

Till original end date

till 2014

Europe

Israel

-15.487704

-47.5744

-0.8710228

-4.433223176

-58.44511

38.03298616

-1.5515038

0.598881863

Armenia

163.612103

372.9322

8.69429131

13.7037181

141.859702

416.4545967

2.68540901

2.462382221

Austria

-70.951157

-35.2418

-6.2979658

-3.567474209

-27.419324

-35.68442529

-0.6243851

-0.704756189

Belgium

-52.49484

-42.7583

-4.2512488

-2.268456889

-73.478493

-27.53215009

-2.990902

-0.733661911

Croatia

58.0304142

358.4954

1.11795453

1.286251058

30.8538892

167.9831859

0.113322

0.820553396

Czech Republic

219.48253

354.8966

1.2450318

1.319071677

71.8918104

137.4748208

0.13263946

0.430620959

Denmark

-77.044882

-67.3123

-6.5670572

-4.506275971

-49.902381

-28.5824473

-1.2639716

-0.598098111

Estonia

672.974324

23.34052

30.3178296

10.44809687

5877.56501

-18.69218172

26.9468342

-3.7466784

Finland

20.4165499

-72.4766

0.24769358

-2.147564674

-56.394189

267.7980926

-0.5329263

0.869948664

France

-72.078316

-54.8002

-17.765674

-8.343797348

-64.338079

-52.11917974

-4.2297913

-2.552064024

Germany

-50.007259

-21.5624

-4.05827

-1.505719388

-54.391227

-7.235007127

-1.555026

-0.121621567

Hungary

-61.63948

62.94649

-0.8315576

0.920062069

202.779037

-46.58784365

0.66571424

-0.085026494

Ireland

-57.282171

-33.0806

-2.4166132

-1.395599131

-49.09711

-40.53821319

-1.0342822

-0.853980074

Italy

-73.087469

-38.9545

-21.709634

-8.634378772

-65.876717

-31.68357733

-5.3032354

-1.813336367

Latvia

1010.39974

125.935

52.691637

38.44257484

2978.76295

226.577343

25.0937467

21.06271165

Luxembourg

-6.6933227

-30.0032

-0.2420114

-3.640261633

-65.324394

71.96361032

-4.7687224

2.59241704

Malta

104.843311

226.9867

9.9035685

6.529178756

NA

0

NA

0

Netherlands

-81.363135

-52.4192

-9.1774405

-2.937934747

-78.81623

-27.14368883

-1.896482

-0.382183675

Norway

-54.65209

16.25017

-1.5919129

0.369758084

-52.487327

40.19554806

-0.7554475

0.29546542

Poland

-16.988135

-16.3808

-0.6362699

-0.854258278

-37.889664

57.96033824

-0.6202435

0.58229383

Romania

-53.034392

49.76146

-15.058785

4.639599921

-72.095933

-11.5163712

-15.719514

-0.894786242

Slovakia

-50.316827

-11.0797

-0.2968504

-0.036522549

NA

0

NA

0

Slovenia

-74.3139

74.30681

-4.0526635

1.138381278

NA

-15.25812121

NA

-0.154280566

Spain

-81.886941

-55.6628

-46.294251

-21.70892456

-79.504323

-42.16429409

-12.375399

-4.05710629

Sweden

-79.216154

-79.1437

-4.2459366

-2.730988032

-25.830436

-72.84339115

-0.4029857

-1.216679299

Switzerland

-79.812558

-50.7581

-16.86722

-4.397688068

-70.382455

-30.65581098

-5.8618442

-1.090489924

United Kingdom

-27.052821

-34.4534

-1.2821637

-1.833181027

-33.998134

-59.18575408

-0.5577417

-1.52031717

Turkey

60.7375928

50.09425

0.67695697

0.55833055

41.4937292

90.26707893

0.07033244

0.153003936

Georgia

5392.65563

1802.003

22.4566197

17.87616985

1123.0204

651.6733187

5.28699864

3.067972726

Kyrgyzstan

6998.62603

174.6448

29.2967188

23.17454332

962.019362

1795.702254

11.1708694

13.40965193

Lithuania

436.498759

75.81398

9.79859263

5.109968787

576.777875

63.87111551

3.02837842

1.097819589

Republic of Moldova

871.796197

33.45943

25.2571731

8.92386586

1467.70457

70.74757727

16.897346

5.588781757

Serbia

-70.513106

11.46153

-7.1856834

0.286442763

-85.069813

-74.35344765

-3.3670369

-0.732947238

Cyprus

19.5314302

-53.5807

0.41938218

-4.599745752

NA

-15.46863826

NA

-0.713063589

Western

Japan

-19.049726

-35.6009

-0.0810955

-0.267776783

-81.990697

-0.0432447

Pacific

-66.35441907

-0.061612907

Malaysia

-7.9750421

-7.97504

-2.1266328

-2.126632796

156.711119

156.7111189

2.31193084

2.311930837

Singapore

-19.736314

32.537

-2.8258231

4.658611157

50.1663088

-49.98741382

0.34365906

-0.342433559

Australia

-65.482964

-41.0907

-6.4895631

-2.602764945

-93.802752

-89.59929366

-1.040554

-0.731103059

 New Zealand

-70.281577

5.776782

-3.898996

0.177032834

-64.49399

-80.75952302

-0.6024464

-1.392127231

Eastern

Egypt

-63.621749

145.0712

-0.2687721

0.10764541

114.383127

0.06181155

Mediterranean

1277.105491

0.382081925

Bahrain

-49.946845

117.8704

-11.1005

4.832645427

-74.240858

-74.17940834

-4.6159756

-4.601178666

Jordan

114.482031

-67.3907

0.81437703

-0.479389012

NA

0

NA

0

Kuwait

33.3174935

-27.6171

0.12045845

-0.560839028

NA

8.614168518

NA

0.048932466

SouthEast Asia

Thailand

-69.968054

1.878263

-192.07258

1.825288073

-73.989224

-0.984658714

-118.2759

-0.55083728

Africa

Mauritius

436.026377

91.03361

109.680758

63.06474674

587.089308

110.4850474

26.6741821

15.75155015

South Africa

56.641946

58.10283

168.964157

173.3219921

25.2653484

29.66264822

78.8817738

92.61072784

Round 3

Reviewer 1 Report

My previous comments have been fully addressed - job well done. I thank the authors for taking the time to add AC alongside PC. After seeing the analysis using uniform start and stop dates, I understand why the authors did not want to include it.

One small typo:

- ln 392. should be "... all databases studied."